# Metabolic Syndrome and Its Related Factors among Hospital Employees: A Population-Based Cohort Study

**DOI:** 10.3390/ijerph18189826

**Published:** 2021-09-17

**Authors:** Yi-Syuan Wu, Wen-Chii Tzeng, Chi-Ming Chu, Wei-Yun Wang

**Affiliations:** 1Graduate Institute of Life Science, National Defense Medical Center, Taipei 11490, Taiwan; pu1254@gmail.com; 2School of Nursing, National Defense Medical Center, Taipei 11490, Taiwan; wctzeng@mail.ndmctsgh.edu.tw; 3School of Public Health, National Defense Medical Center, Taipei 11490, Taiwan; chuchiming@web.de; 4Department of Nursing, Tri-Service General Hospital and School of Nursing, National Defense Medical Center, Taipei 11490, Taiwan

**Keywords:** hospital employees, metabolic syndrome, unhealthy behaviours, body mass index, biochemical markers

## Abstract

Several studies have reported on metabolic syndrome (MetS) based on cross-sectional designs, which cannot show a long-term result. Information is lacking on MetS and related factors based on a longitudinal cohort. This study aimed to examine the relationship between MetS and related factors for a total of six years among hospital employees. A population-based study was conducted, including 746 staff. A total of 680 staff without MetS in 2012 were enrolled in the analysis for repeated measurement of six years of the longitudinal cohort. Data were retrieved from the hospital’s Health Management Information System. Analyses were performed using Student’s *t*-test, chi-square test, logistic regression, and generalised estimating equations. Statistical significance was defined as *p* < 0.05. Hospital employees aged between 31 and 40 (odds ratio (OR) = 4.596, *p* = 0.009), aged between 41 and 50 (OR = 7.866, *p* = 0.001), aged greater than 50 (OR = 10.312, *p* < 0.001), with a body mass index (BMI) of 25.0~29.9 kg/m^2^ (OR = 3.934, *p* < 0.001), a BMI ≥ 30 kg/m^2^ (OR = 13.197, *p* < 0.001), higher level of white blood counts (β = 0.177, *p* = 0.001), alanine aminotransferase (β = 0.013, *p* = 0.002), and uric acid (β = 0.223, *p* = 0.005) were at risk of being diagnosed with MetS. The identification of at-risk hospital employees and disease management programs addressing MetS-related factors are of great importance in hospital-based interventions.

## 1. Introduction

Employees are an important part of any institution. Since the promulgation of the Ottawa Charter for Health Promotion by the World Health Organization (WHO, Geneva, Switzerland) in 1986, global support for workplace health promotion has encouraged employers to ensure employee health [1]. Employee health is related to an employee’s ability to work because diseases or health problems can prevent employees from completely engaging in work activities and diminish their workload [2,3]. The estimated prevalence of metabolic syndrome (MetS) is 9.5% in machine operators and 7.5–10.6% in Dutch trade workers [4]. In Taiwan, 7.7% of labourers [5], 8.2% of high-tech labour [1], and 10.3% of hospital employees [6] meet the criteria for metabolic syndrome. According to the Health Promotion Hospital (HPH) trend, the health of hospital employees is a matter of great concern for hospital management. Hospitals should provide health care services to the public and patients and focus on hospital employees to make the health care. Although hospital employees have better health knowledge, high-pressure work may threaten their health. Moreover, MetS increase individual health care costs by US$ 3719, which may have a major impact on the fiscal performance of an institution [1]. Therefore, the prevention of MetS is an important issue for hospital employees.

MetS is a cluster of metabolic abnormalities that include central obesity, high blood pressure, atherogenic dyslipidaemia, and insulin resistance, increasing the risks of cardiovascular diseases, diabetes mellitus, and chronic kidney diseases [7]. MetS also contributes to all-cause and cardiovascular disease-related mortality [8]. Therefore, an institution must evolve a more holistic, total population health management (PHM) approach to identify high-risk individuals of MetS early and promote employee health [2,9,10]. There are multiple risk factors for MetS, such as demographic characteristics, e.g., being male [11,12,13] or a middle-aged or older adult [14,15]. In hospital health care settings, work schedules are commonly characterized by alternating shifts, which make hospital employees particularly vulnerable, as their jobs require them to work night or rotating shifts. The conflicting time signals given by variations in exposure to light at night are proposed to cause the loss of internal synchrony. The causes of adverse metabolic outcomes have been posited to include a causal link between metabolic regulation and the molecular function of the circadian rhythm [16]. Previous research indicated that individual characteristics, such as shift work, chronic disease, family history [17], higher body mass index (BMI) [13,18], and unhealthy behaviours [19,20,21], are associated with higher rates of MetS. In summary, past studies pointed out that being male, being a middle-aged or older adult, having chronic diseases, having greater family histories, participating in shift work, smoking cigarettes, drinking alcohol, and chewing betel nuts were correlated with MetS [11,12,13,14,15,16,17,18,19,20,21]. In addition, some healthy behaviours, such as eating vegetables and fruits, drinking milk, and brushing teeth, were not mentioned in past studies of MetS. Therefore, this study considered the relationship of healthy behaviours with MetS.

Previous studies have indicated that prevention and intervention in the early stages effectively reduce the severity of and prevent cardiovascular disorders [22,23]. However, most of the studies related to MetS have had a cross-sectional design. Few studies have discussed MetS and its related factors among hospital employees with a longitudinal cohort. Therefore, the purpose of this study was to examine the relationship between MetS and its related factors for a total of six years among hospital employees.

## 2. Materials and Methods

### 2.1. Study Design

This study was designed as a population-based and prospective longitudinal cohort study. This study was approved by the Institutional Review Board of the Tri-Service General Hospital in Taiwan (approval number: 2-107-05-030).

### 2.2. Data Source

Data were retrieved from the Health Management Information System (HMIS, Tri-Service General Hospital, Taipei, Taiwan), established in 2012 at the hospital. We used data from annual worksite health check-ups for employees in the hospital. The database included demographic information (i.e., sex and date of birth), work type (i.e., shift work), healthy behaviours (i.e., drinking milk, eating vegetables and fruits, and brushing teeth), unhealthy behaviours (i.e., smoking, alcohol consumption, and chewing betel nut), past history (i.e., number of chronic diseases and number of family history), physical measurements (i.e., height, weight, waist circumstance (WC), blood pressure (BP), and pulse rate), and the biochemical profile (i.e., white blood count (WBC), haemoglobin (Hb), fasting plasma glucose (FPG), cholesterol, triglyceride (TG), aspartate aminotransferase (AST), alanine aminotransferase (ALT), blood urea nitrogen (BUN), total calcium, phosphorus, uric acid, creatinine (Cr), alkaline phosphatase, total bilirubin, total protein, albumin, albumin to globulin ratio (A/G ratio), HDL-C, low-density lipoprotein cholesterol (LDL-C), and the atherogenic index of plasma (AIP).

Demographic information, work type, healthy behaviours, unhealthy behaviours, and past history were recorded with a self-report questionnaire. Height and weight were measured using a regularly corrected height and weight metre. WC was measured using an inelastic tape at the umbilicus level at the end of exhalation while the individual was standing stably on both feet. Systolic and diastolic BP (SBP and DBP, respectively) and pulse rate were measured using an automatic sphygmomanometer and recorded after the individual had rested for at least 5 min. BMI was calculated by dividing weight (in kilograms) by height (in square metres). Overweight was defined as a BMI between 25 and 29.9 kg/m^2^, whereas a BMI of ≥30 kg/m^2^ indicated obesity. Biochemical profiles were measured from venous blood samples, which were collected after an overnight fast. Serum biochemical markers were measured using a TBA-c16000 automated analyser (Toshiba Medical Systems, Tochigi, Japan). AIP was calculated as the logarithmic transformation of the ratio of TG to HDL-C and was calculated according to the following formula: AIP = Log [TGs]/[HDL-C] [24]. AIP values of −0.3 to 0.1 are associated with low risk, 0.1–0.24 with medium risk, and above 0.24 with a high risk of CVD [25]. All identification numbers in the database are encrypted to ensure individuals’ privacy.

### 2.3. Study Population

A total of 1868 hospital employees completed the health examination in 2012. A total of 1122 employees who did not complete six years of health check-ups were excluded. A total of 746 hospital employees who completed an annual health check-up during 2012–2017 were included in this study for MetS and non-MetS, and their demographic characteristics, physical characteristics, and biochemical profile were compared. The study also measured the association between MetS and demographic information, work type, healthy behaviours, unhealthy behaviours, past history, physical measurements, and biochemical profile by repeated measurement rather than one-shot measurement. Thus, we selected 680 individuals without MetS in 2012 who were enrolled in the analysis for six repeated measurements. After six years of follow-up, 601 hospital employees were not diagnosed with MetS, and 79 were diagnosed with MetS. The flow chart of the enrolment of study participants is shown in Figure 1.

MetS was diagnosed based on the NCEP-ATP III guidelines and modified criteria from the International Diabetes Federation’s definition accounting for WC norms in the Asian population, that is, based on ≥3 of any of the following 5 abnormalities: (1) central obesity (WC ≥ 80 cm in women or ≥90 cm in men); (2) high BP (SBP ≥ 130 mmHg or DBP ≥ 85 mmHg); (3) hyperglycaemia (FPG level ≥ 100 mg/dL); (4) hypertriglyceridaemia (TG level ≥ 150 mg/dL); and (5) low HDL-C level (<50 mg/dL in women or <40 mg/dL in men) [26].

### 2.4. Statistical Analysis

Data management and statistical analyses were performed using SPSS (Version 27.0, SPSS Inc., Chicago, IL, USA). First, participants were categorized into 2 groups according to the presence or absence of MetS. The demographics, work type, health behaviours, unhealthy behaviours, past history, physical measurements, and biochemical result distributions of the MetS and non-MetS groups were compared. The student’s *t*-test was used for continuous variable comparisons between groups, whereas the chi-square test was used for categorical variables. We used logistic regression to identify variables with a significance level of *p* < 0.05. To take the within-subject correlation into account, generalised estimating equations (GEEs) with a logit link function were conducted to estimate the relative risks of demographics, work type, past history, unhealthy behaviours, physical measurements, and biochemical factors at the second stage.

## 3. Results

### 3.1. Comparison of Participant Characteristics by the Presence of Metabolic Syndrome

Data from 2012 to 2017 on a total of 746 staff members were included in this study; 226 were male, and 520 were female. The mean age was 38.40 years (standard deviation = 9.10). The majority of participants were aged between 31 and 40 years. Age (*t* = −4.23, *p* < 0.001), age categories (χ^2^ = 15.66, *p* = 0.001), shift work (χ^2^ = 6.24, *p* = 0.013), smoking status (χ^2^ = 6.31, *p* = 0.012), chewing betel nut (χ^2^ = 6.15, *p* = 0.013), number of chronic diseases (χ^2^ = 19.27, *p* < 0.001), BMI (*t* = −13.24, *p* < 0.001), and BMI categories (χ^2^ = 147.79, *p* < 0.001) were significantly different between groups with and without metabolic syndrome. Participants’ characteristics between the groups with and without metabolic syndrome are shown in Table 1.

### 3.2. Physical and Biochemical Values of the Participants

The average weight, WC, systolic BP, diastolic BP, WBC, haemoglobin, FPG, cholesterol, TG, AST, ALT, total calcium, uric acid (UA), alkaline phosphatase, A/G ratio, and HDL-C were significantly different between the groups with and without MetS. People with MetS had significantly higher weight (*t* = −10.48, *p* < 0.001), higher WC (*t* = −13.13, *p* < 0.001), higher SBP (*t* = −11.73, *p* < 0.001), higher DBP (*t* = −9.24, *p* < 0.001), higher WBC (*t* = −4.34, *p* < 0.001), higher haemoglobin (*t* = −3.29, *p* = 0.001), higher FPG (*t* = −4.48, *p* < 0.001), higher cholesterol (*t* = −2.80, *p* = 0.005), higher triglycerides (*t* = −6.48, *p* < 0.001), higher AST (*t* = −2.13, *p* = 0.033), higher ALT (*t* = −3.65, *p* < 0.001), higher total calcium (*t* = −2.22, *p* = 0.027), higher uric acid (*t* = −4.87, *p* < 0.001), higher alkaline phosphatase (*t* = −3.60, *p* < 0.001), lower A/G ratio (*t* = 2.07, *p* = 0.039), lower HDL-C (*t* = 12.75, *p* < 0.001), higher LDL-C (*t* = −2.69, *p* = 0.008), and higher AIP (*t* = −12.96, *p* < 0.001). Detailed data are summarized in Table 2.

### 3.3. Risk Factors for Metabolic Syndrome in Hospital Employees

The risk factors related to MetS of hospital employees are shown in Table 3. After controlling for sex, shift work, number of chronic diseases, number of family history, smoking and alcohol status, age, BMI, WBC, ALT, and uric acid were significantly associated with a higher risk of MetS. Hospital employees aged between 31 and 40 (OR = 4.596, *p* = 0.009), those aged between 41 and 50 (OR = 7.866, *p* = 0.001), those aged greater than 50 (OR = 10.312, *p* < 0.001), those with a BMI 25.0~29.9 kg/m^2^ (OR = 3.934, *p* < 0.001), and those with a BMI ≥ 30 kg/m^2^ (OR = 13.197, *p* < 0.001) had a significantly elevated risk of MetS. Regarding biochemical values, a one-unit increase in WBC count increased the OR for MetS by 19.4% (β = 0.177, *p* = 0.001). A one-unit increase in ALT or UA was associated with an increase of 1.3% (β = 0.013, *p* = 0.002) and 25% (β = 0.223, *p* = 0.005), respectively, in the risk of being diagnosed with MetS.

## 4. Discussion

This study is one of the few to have investigated MetS and its related factors among hospital employees with a longitudinal cohort. The results of the GEE model showed that increasing age, higher BMI, and higher WBC, ALT, and UA levels were associated with a significantly elevated risk of MetS, which is similar to previous findings [1,6,21,27,28]. However, Ho et al., Shafique et al., and Rhee et al. used a cross-sectional design [6,20,28]. Our study utilized a comprehensive longitudinal cohort study to link the risk factors and development MetS. This indicates that all risk factors might play a role in the causation of MetS, especially concerning increased age and elevated levels of BMI, WBC, ALT, and UA, which might be a clinical pathway for developing a diagnostic component for MetS.

Our findings showed that age was correlated with the risk of MetS, which is true for all other population studies [14,16,29]. This is not surprising since there are many commonalities in the biochemical changes of the ageing process [29]. It can be inferred that blood vessels gradually lose elasticity and gain resistance with age, slowing blood flow. Moreover, with poor circulation, fat is prone to accumulate in the abdomen and release free fatty acids into the serum, leading to higher insulin resistance, elevated serum triglyceride levels, increased LDL-C levels, and, consequently, a greater risk of MetS [1]. Our study also showed that higher BMI was associated with greater OR for developing MetS, confirming previous reports [1,30,31]. BMI can be used to assess body fat content, which indicates excessive weight gain. In cases of excessive weight gain, adipose cells release free fatty acids and tumour necrosis factor-α (TNF-α), which blocks phosphatidylinositide-3-kinase-dependent signal transduction pathways and inhibits the insulin-based regulation of the uptake and utilization of glucose. Eventually, it reduces glucose uptake in the liver and skeletal muscles and induces hyperglycaemia and insulin resistance, leading to metabolic abnormalities [1,17].

We further noted that subjects who had a habit of chewing betel nuts were borderline associated with a greater OR for developing MetS, confirming a previous finding [20]. The hydroxychavicol, areca alkaloids, and arecoline compounds found in areca nut affect metabolic processes, which may inhibit adipose tissue differentiation and induce adenylyl-cyclase-dependent lipolysis and interfere with the insulin signalling pathway associated with glucose uptake, all of which contribute to hyperlipidaemia and insulin resistance. At the same time, areca alkaloids also stimulate appetite via inhibition of Gamma-Aminobutyric Acid (GABA) receptors; an established oral chewing habit can also increase appetite, contributing to the risk of obesity or insulin resistance. In addition, areca nut-specific nitrosamines might promote glucose intolerance due to central obesity. Moreover, areca nut may induce reactive oxygen species (ROS) production, cause cell cycle aberrations and irregular cell differentiation, induce platelet aggregation, and increase the lipid profile burden by increasing oxidative stress and inflammation [19,20]. In this way, chewing betel nut increases the risk of MetS.

We also observed that our participants with MetS had increased WBC counts. Similar results have been reported by international studies [1]. An elevated WBC count indicates subclinical systemic chronic inflammation. In an inflammatory state, vascular endothelial cells secrete proinflammatory cytokines, such as interleukin-6 or TNF-α, which can activate and attract many WBCs to the damaged region within the vessel’s lumen. Following infiltration into the tunica media, WBCs absorb oxidized LDL-C and become foam cells. This process may greatly increase the risk of vascular disease [1,28]. Another potential pathogenic mechanism was Rho kinase, which WBCs secrete. Liu et al. found that Rho-kinase can block insulin signalling by stimulating the phosphorylation of insulin receptor substrate-1. This action prevents the translocation of glucose to the cell membrane by glucose transporter 4, thereby leading to insulin resistance and MetS [1,28].

In the current study, ALT was a significant factor for MetS, even after controlling for potential confounders. Another study also showed that elevated ALT can be frequently observed among patients with MetS [28]. A growing body of evidence supports the association of elevated ALT with tremendous workplace pressure or extreme fatigue. Once work pressure increases, so do the prevalence of MetS [1,6,32]. Otherwise, elevated ALT was also associated with fatty liver disease owing to fatness. Research indicates that once fatty liver disease occurs, widespread lipid accumulation in muscles and other organs, such as the pancreas, facilitates insulin resistance and increases susceptibility to MetS [1,6,32].

The present study results revealed that employees with MetS have a higher risk of elevated UA, which is consistent with the findings of a high-tech industry-based survey conducted in Taiwan [1]. UA, the end product of purine metabolism, has been reported to be linked with hypertension, insulin resistance, and atherosclerosis. Possible mechanisms linked to elevated serum UA levels include inhibiting the production of nitric oxide (NO) by endothelial cells. NO can serve as an inhibitor of thrombosis and inflammatory reactions, further preventing the growth and transfer of vascular smooth muscle cells [1,28]. Therefore, higher UA levels and MetS were proposed to be associated with oxidative stress, endothelial dysfunction, and inflammation.

Some factors, such as total calcium, shift work, and smoking status, were found to be related to the risk of MetS in our study. Calcium levels were positively associated with insulin resistance, and impaired glucose metabolism resulted in high glucose concentrations. Otherwise, increased serum calcium levels lead to calcium influx into arterial smooth muscles via calcium channels, increasing cytosolic calcium and inducing muscle contraction and arterial vasoconstriction. As a result, increased BP and peripheral vascular resistance can be observed [33]. In particular, high levels of BP and glucose were the diagnostic criteria of MetS. Therefore, our results indicate that among patients with MetS, calcium levels appear to play an important role as a differentiating factor. However, the effect of calcium levels on MetS remains poorly established [34]. Further studies will be necessary to better address this topic. Shift and night work appears to have a negative effect on hospital employee health, possibly due to its effect on sleep deprivation, circadian desynchronization, high levels of stress, and behavioural changes in diet and physical activity. These factors could affect glucose tolerance and develop into a subclinical hypercortisol state, which may induce a decrease in HDL-C with increases in TG and FPG [16,17,35,36]. Prolonged exposure to these factors may also entrain the cardiovascular system to operate at an elevated pressure equilibrium through structural adaptations such as left ventricular hypertrophy and may significantly affect hospital employee health [17,36].

In addition, the studies of Ho et al. [6] and Tsai et al. [1] did not collect data on smoking status, alcohol consumption, or haemoglobin levels, which are potential confounding variables for MetS that might elicit inferential error in biochemical markers. In this way, our study overcame these limitations by including the variables of smoking status, alcohol consumption, and haemoglobin levels and utilizing the longitudinal cohort method to examine the relationships between these variables and MetS. We discovered that haemoglobin was associated with MetS, as determined by the t-test. However, the opposite was true in the GEE model. One possible explanation is that the abnormally high haemoglobin concentration might lead to high blood viscosity and increased vascular occlusion, causing fat and total cholesterol retention within the cells and ultimately resulting in heart disease or stroke [1,6]. Because the specific biological mechanisms by which haemoglobin levels increase the risk of MetS remain unclear [1], further study is needed to clarify the effect of haemoglobin levels on MetS.

Drinking milk and eating vegetables and fruits were not discovered to be related to the risk of MetS in our study. This may be because Asian dietary components are full of soy foods and are particularly abundant in isoflavones [37]. According to the study of Chen et al. [38], isoflavones could affect the gut microbiome. These effects appear to be associated with regulating the diversity and structure of the intestinal microbiota and effectively resolving abnormal serum lipid levels. Peng et al. [39] also found that isoflavones can improve glucose, lipid, and blood pressure control. Therefore, habitual intake of isoflavones is inversely associated with the risk of MetS and its components [40]. However, few studies have discussed the relationships among dietary components, isoflavones, the gut microbiome, and MetS in hospital employees. Further study is needed to investigate these issues.

This study has several limitations. First, the HMIS contains limited information on potentially confounding factors (e.g., lifestyle patterns, physical activity, medication history). It lacks a subdivision of dietary patterns, such as Mediterranean-style diet, plant-based diet, and Nordic dietary patterns, which restrict the findings of this study. Second, the dataset contains self-reported information, which may carry some bias, such as recall bias. Third, the results cannot be generalized because this study focused on only one hospital’s employees. Although we utilized a six-year longitudinal cohort to follow the sample, our participants included considerably fewer men. Additional studies should be conducted in a clinical setting to overcome these limitations and investigate hospital employees’ lifestyle patterns and physical activity.

## 5. Conclusions

Our study improves the understanding of MetS among hospital employees with data from a long-term follow-up hospital health examination. The results of the longitudinal study design showed that increasing age, higher BMI, and higher WBC, ALT, and UA levels were factors that had a significantly elevated risk of MetS over the six years. Risk factors for MetS in hospital employees were identified using a health examination. In summary, identifying at-risk hospital employees and disease management programs addressing MetS-related factors are of great importance in hospital-based interventions. Employers should focus on reducing risk factors for developing MetS among hospitals.

## Figures and Tables

**Figure 1 ijerph-18-09826-f001:**
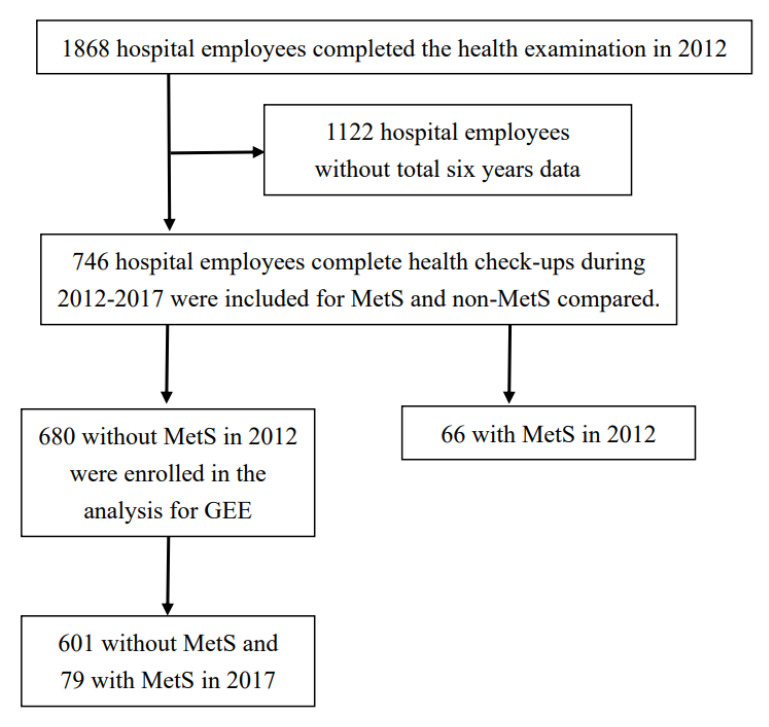
Flow chart of enrolment of study participants.

**Table 1 ijerph-18-09826-t001:** Characteristics of hospital staff with and without metabolic syndrome.

Variables	Total (*n* = 746)	Metabolic Syndrome		χ^2^/t	*p*
		No (*n* = 680)	Yes (*n* = 66)		
Age (years)	38.40 ± 9.10 ^a^	37.97 ± 9.00 ^a^	42.88 ± 8.96 ^a^	−4.23	<0.001
Age categories (years)				15.66	0.001
>50	99	83 (12.2)	16 (24.2)		
41–50	177	156 (22.9)	21 (31.8)		
31–40	308	284 (41.8)	24 (36.4)		
21–30	162	157 (23.1)	5 (7.6)		
Sex				1.26	0.261
Male	226	202 (29.7)	24 (36.4)		
Female	520	478 (70.3)	42 (63.6)		
Shift work				6.24	0.013
Yes	587	543 (79.9)	44 (66.7)		
No	159	137 (20.1)	22 (33.3)		
Drinking milk (*n* = 731) ^b^				1.76	0.185
Yes	158	140 (21.0)	18 (28.1)		
No	573	527 (79.0)	46 (71.9)		
Eating at least three servings of vegetables and two of fruits (*n* = 729) ^b^				0.03	0.867
Yes	289	263 (39.5)	26 (40.6)		
No	440	402 (60.5)	38 (59.4)		
Tooth brushing (*n* = 727) ^b^				2.83	0.243
One time/day	118	104 (15.7)	14 (21.9)		
Two times/day	473	431 (65.0)	42 (65.6)		
More than three times/day	136	128 (19.3)	8 (12.5)		
Smoking status (*n* = 732) ^b^				6.31	0.012
Yes	41	33 (4.9)	8 (12.5)		
No	691	635 (95.1)	56 (87.5)		
Alcohol (*n* = 731) ^b^				1.66	0.198
Yes	244	218 (32.7)	26 (40.6)		
No	487	449 (67.3)	38 (59.4)		
Chewing betel nut (*n* = 731) ^b^				6.15	0.013
Yes	5	3 (0.4)	2 (3.1)		
No	726	664 (99.6)	62 (96.9)		
Number of chronic diseases				19.27	<0.001
≥2	38	28 (4.1)	10 (15.2)		
1	192	170 (25.0)	22 (33.3)		
0	516	482 (70.9)	34 (51.5)		
Number of family history				7.92	0.095
≥4	28	24 (3.5)	4 (6.1)		
3	66	56 (8.2)	10 (15.2)		
2	150	133 (19.6)	17 (25.8)		
1	205	189 (27.8)	16 (24.2)		
0	297	278 (40.9)	19 (28.8)		
BMI (kg/m^2^)	23.54 ± 3.91 ^a^	22.96 ± 3.49 ^a^	28.97 ± 3.82 ^a^	−13.24	<0.001
BMI categories (kg/m^2^)				147.86	<0.001
≥30.0	50	27 (4.0)	23 (34.8)		
25.0–29.9	183	147 (21.6)	36 (54.5)		
18.5–24.9	473	467 (68.7)	6 (9.1)		
<18.5	40	39 (5.7)	1 (1.5)		

Values are counts (percentages) unless stated otherwise, ^a^ Mean ± standard deviation, ^b^ There are some missing values, abbreviations: BMI, body mass index.

**Table 2 ijerph-18-09826-t002:** Physical and biochemical values of the study population.

Variables	Reference Value	Mean ± Sd (*n* = 746)	Metabolic Syndrome Mean ± Sd		*t*	*p*-Value
			No (*n* = 680)	Yes (*n* = 66)		
Height	-cm	163.12 ± 7.97	163.11 ± 7.97	163.24 ± 7.99	−0.13	0.896
Weight	-kg	62.92 ± 12.71	61.50 ± 11.77	77.54 ± 12.89	−10.48	<0.001
WC	female < 80 cm male < 90 cm	75.92 ± 10.40	74.52 ± 9.29	90.39 ± 10.20	−13.13	<0.001
SBP	<120 mmHg	115.69 ± 14.04	113.96 ± 12.63	133.48 ± 15.53	−11.73	<0.001
DBP	<80 mmHg	70.92 ± 10.70	69.65 ± 9.64	83.98 ± 12.25	−9.24	<0.001
Pulse rate	60–80/min	78.34 ± 10.72	78.11 ± 10.52	80.68 ± 12.44	−1.86	0.063
WBC	4.5–11 × 10^3^/uL	6.36 ± 1.69	6.26 ± 1.63	7.38 ± 2.03	−4.34	<0.001
Haemoglobin	female 12.0~16.0 male 13.5~18.0 gm/dL	13.67 ± 1.54	13.62 ± 1.51	14.27 ± 1.74	−3.29	0.001
FPG	70~100 mg/dL	92.53 ± 18.14	90.49 ± 12.17	113.55 ± 41.61	−4.48	<0.001
Cholesterol	<200 mg/dL	192.62 ± 34.81	191.52 ± 34.62	204.05 ± 34.91	−2.80	0.005
Triglycerides	<150 mg/dL	99.38 ± 61.51	92.01 ± 50.11	175.41 ± 103.43	−6.48	<0.001
AST (GOT)	8~31 U/L	19.79 ± 12.51	19.49 ± 12.52	22.92 ± 12.09	−2.13	0.033
ALT (GPT)	0~41 U/L	21.57 ± 24.24	20.58 ± 24.07	31.77 ± 23.77	−3.65	<0.001
BUN	7~25 mg/dL	12.53 ± 3.37	12.47 ± 3.36	13.11 ± 3.47	−1.46	0.144
Total calcium	8.6~10.2 mg/dL	9.37 ± 0.34	9.36 ± 0.34	9.46 ± 0.35	−2.22	0.027
Phosphorus	2.7~4.5 mg/dL	3.70 ± 0.49	3.71 ± 0.49	3.66 ± 0.53	0.84	0.400
Uric acid	2.3~7.0 mg/dL	5.33 ± 1.47	5.25 ± 1.44	6.16 ± 1.50	−4.87	<0.001
Creatinine	0.5~0.9 mg/dL	0.75 ± 0.17	0.74 ± 0.16	0.76 ± 0.19	−0.61	0.546
Alkaline phosphatase	35~104 U/L	60.99 ± 16.94	60.29 ± 16.62	68.11 ± 18.66	−3.60	<0.001
Total bilirubin	0.3~1.0 mg/dL	0.61 ± 0.27	0.61 ± 0.27	0.60 ± 0.26	0.41	0.682
Total protein	6.6~8.7 g/dL	7.43 ± 0.37	7.42 ± 0.37	7.51 ± 0.34	−1.95	0.051
Albumin	3.5~5.7 g/dL	4.70 ± 0.25	4.70 ± 0.25	4.69 ± 0.22	0.44	0.659
A/G ratio	1.2~2.4	1.75 ± 0.24	1.76 ± 0.25	1.69 ± 0.23	2.07	0.039
HDL-C	>65 mg/dL	69.34 ± 16.56	65.56 ± 16.14	47.23 ± 10.55	12.75	<0.001
LDL-C	<120 mg/dL	113.40 ± 29.87	112.11 ± 29.45	120.59 ± 31.28	−2.69	0.008
AIP	−0.3~0.1	0.15 ± 0.29	0.11 ± 0.27	0.53 ± 0.25	−12.96	<0.001

Abbreviations: Sd, standard deviation; WC, waist circumference; SBP, systolic blood pressure; DBP, diastolic blood pressure; WBC, white blood cell; FPG, fasting plasma glucose; AST, aspartate aminotransferase; ALT, alanine aminotransferase; BUN, blood urea nitrogen; A/G ratio, albumin globulin ratio; HDL-C, high-density lipoprotein cholesterol; LDL-C, low-density lipoprotein cholesterol; AIP, atherogenic index of plasma.

**Table 3 ijerph-18-09826-t003:** Risk factors for metabolic syndrome in hospital employees.

Variable	Beta (95% CI)	*p*-Value	Odds Ratio (95% CI)
Sex			
Male	0.019 (−0.509~0.547)	0.943	1.020 (0.601~1.729)
Female			
Shift work			
Yes	0.059 (−0.410~0.527)	0.806	1.060 (0.664~1.694)
No			
Number of chronic diseases			
≥2	0.036 (−0.557~0.628)	0.906	1.036 (0.573~1.874)
1	−0.080 (−0.460~0.300)	0.680	0.923 (0.631~1.350)
0			
Number of family history			
≥4	0.373 (−0.260~1.006)	0.248	1.452 (0.771~2.734)
3	0.235 (−0.271~0.741)	0.363	1.265 (0.763~2.097)
2	0.133 (−0.590~0.325)	0.570	0.876 (0.554~1.384)
1	−0.217 (−0.654~0.220)	0.331	0.805 (0.520~1.247)
0			
Smoking status			
Yes	−0.404 (−1.259~0.450)	0.354	0.667 (0.284~1.569)
No			
Alcohol			
Yes	−0.185 (−0.612~0.243)	0.398	0.831(0.542~1.275)
No			
Age categories (years)			
>50	2.333 (1.126~3.541)	<0.001	10.312 (3.083~34.493)
41–50	2.063 (0.877~3.248)	0.001	7.866 (2.405~25.729)
31–40	1.525 (0.373~2.677)	0.009	4.596 (1.452~14.544)
21–30			
BMI categories (kg/m^2^)			
≥30	2.580 (2.009~3.151)	<0.001	13.197 (7.459~23.351)
25.0–29.9	1.370 (0.906~1.834)	<0.001	3.934 (2.474~6.256)
<24.9			
Chewing betel nut			
Yes	1.260 (−0.140~2.661)	0.078	3.526 (0.869~14.307)
No			
White blood cell	0.177 (0.074~0.280)	0.001	1.194 (1.077~1.324)
Alanine aminotransferase	0.013 (0.005~0.021)	0.002	1.013 (1.005~1.021)
Uric acid	0.223 (0.069~0.378)	0.005	1.250 (1.072~1.459)
Haemoglobin	0.001 (−0.161~0.163)	0.988	1.001 (0.851~1.177)
Total calcium	0.487 (−0.038~1.012)	0.069	1.628 (0.963~2.751)

Abbreviations: CI, confidence interval; BMI, body mass index.

## Data Availability

Publicly available datasets were analysed in this study.

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
