# Peer review of "Metabolic Syndrome and Its Related Factors among Hospital Employees: A Population-Based Cohort Study"

_ijerph, 2021, doi:10.3390/ijerph18189826_

Round 1
Reviewer 1 Report
In the present paper, Yi-Syuan Wu and coworkers examined the relationship between metabolic syndrome (MetS) and related factors for a total of six years among hospital employees. Specifically, a population-based and longitudinal cohort was conducted including 746 staffs. Data were retrieved from the hospital’s Health Management Information System. The authors concluded that the identification of at-risk hospital employees and disease management programs addressing MetS-related factors are of great importance in hospital-based interventions. Overall, I think that the paper could be of interest for readers and researchers, in general.
I make some suggestions for further improve the quality of the manuscript.
The authors, if possible, should incorporate in tables the dietary pattern of the patients included in the present study (e. Mediterranean-style diet, Plants-based diet, Nordic dietary pattern, etc.); in this way, I feel that the readers can better understand the results obtained in the present clinical study and their possible application to clinical practice.
“Oriental diet” is particularly abundant in isoflavones. So, the overall effects on MetS observed in clinical studies could be related to these phytochemicals and no adherence/not compliance to diet. Please discuss this crucial aspect in the revised manuscript.
Some medicines/drugs can be used to treat MetS. This aspect could interfere with results here revealed. Please discuss this aspect and eventually take this factor into account in statistical analysis.
Recent research suggested that dietary fibers from beans, fruits and vegetables were associated with the gut microbiome composition and, accordingly, with risk of developing MetS. Does the authors plan to assess microbioma composition in their population? Please make a comment in the discussion section of revised manuscript.
Author Response
1
Dear Editor and Reviewers,
Thank you for the opportunity to revise manuscript # ijerph-1324639. We have revised the manuscript based
on the editor and reviewer comments and we wish the quality of the manuscript has improved. Thank you
again for your time and expertise in reviewing this manuscript for publication in Journal of Environmental
Research and Public Health.
Reviewer #1 Comments | Author Response to Comments |
The authors, if possible, should incorporate in tables the dietary pattern of the patients included in the present study (e. Mediterranean-style diet, Plants-based diet, Nordic dietary pattern, etc.); in this way, I feel that the readers can better understand the results obtained in the present clinical study and their possible application to clinical practice. |
Thank you for the comments. Because the “American Heart Association” pointed out “Higher consumption of fruits and vegetables is associated with a lower risk of death in men and women, according to data representing nearly 2 million adults. Five daily servings of fruits and vegetables, eaten as 2 servings of fruit and 3 servings of vegetables, may be the optimal amount and combination for a longer life. These findings support current U.S. dietary recommendations to eat more fruits and vegetables and the simple public health message '5-a-day.'” (retrieval from https://www. sciencedaily.com/releases/2021/03/210301084519. htm) In this way, the self- report questionnaire about the healthy dietary in our research, we only survey the “drinking milk” and “eating at least three servings of vegetables and two of fruits”. In our research, we don’t subdivide the dietary patterns such as Mediterranean-style diet, Plant-based diet, Nordic dietary patterns. This is our research limitation in current. However, based on the reviewer’s comments, we also try to add the variables of “drinking milk” and “eating vegetables and fruits” into the generalized estimating equations (GEEs) model to examine the relationships with metabolic syndrome again. The output of the statistics as below. |
2
From the results of the GEE model, we found that drinking milk, eating vegetables and fruits were not discovered to be related to the risk of metabolic syndrome in our study. This maybe “Oriental diet” is particularly abundant in isoflavones. As suggested, we have addressed in the limitation and recommendation: “the Health Management Information System (HMIS) contains limited information on potentially confounding factors (e.g., lifestyle patterns and physical activity), and lacks subdivision of dietary patterns, such as Mediterranean-style diet, plant-based diet, and Nordic dietary patterns, which restrict the findings of this study”. Please see the page 12 line 7. We can subdivision the dietary patterns to survey the relationships with metabolic syndrome in the future. |
|
“Oriental diet” is particularly abundant in isoflavones. So, the |
Thank you for the comments. We have discussed the relationship between the isoflavones and metabolic syndrome. “Drinking milk |
3
overall effects on MetS observed in clinical studies could be related to these phytochemicals and no adherence/not compliance to diet. Please discuss this crucial aspect in the revised manuscript. |
and eating vegetables and fruits were not discovered to be related to the risk of MetS in our study. This may be because Asian dietary components are full of soy foods and are particularly abundant in isoflavones [38]. According to the study of Chen et al. [39], isoflavones could affect the gut microbiome, and these effects appear to be associated with regulation of the diversity and structure of the intestinal microbiota and effectively resolve abnormal serum lipid levels. Peng et al. [40] also found that isoflavones can improve glucose, lipid, and blood pressure control. Therefore, habitual intake of isoflavones is inversely associated with the risk of MetS and its components [41]. However, few studies have discussed the relationships among dietary components, isoflavones, the gut microbiome, and MetS in hospital employees. Further study is needed to investigate these issues”. Please see the page 11 line 19. |
Some medicines/drugs can be used to treat MetS. This aspect could interfere with results here revealed. Please discuss this aspect and eventually take this factor into account in statistical analysis. |
Thank you for the comments. In our research, we don’t survey the medication history. This is our research limitation now. As suggested, we have addressed in the limitation and recommendation: “the HMIS contains limited information on potentially confounding factors (e.g., lifestyle patterns, physical activity, medication history) and lacks subdivision of dietary patterns, such as Mediterranean-style diet, plant-based diet, and Nordic dietary patterns, which restrict the findings of this study”. Please see the page 12 line 7. We can investigate the relationship between the medicines/drugs with the metabolic syndrome in the future. |
Recent research suggested that dietary fibers from beans, fruits and vegetables were associated with the gut microbiome composition and, accordingly, with risk of developing MetS. Does the authors plan to assess microbioma composition in their population? Please make a comment in the discussion section of revised manuscript. |
Thank you for the comments. We will assess microbioma composition in hospital employees in the future. We have revised as below. “Drinking milk and eating vegetables and fruits were not discovered to be related to the risk of MetS in our study. This may be because Asian dietary components are full of soy foods and are particularly abundant in isoflavones [38]. According to the study of Chen et al. [39], isoflavones could affect the gut microbiome, and these effects appear to be associated with regulation of the diversity and structure of the intestinal microbiota and effectively resolve abnormal serum lipid levels. Peng et al. [40] also found that isoflavones can improve glucose, lipid, and blood pressure control. Therefore, habitual intake of isoflavones is inversely associated with the risk of MetS and its components [41]. However, few studies |
4
have discussed the relationships among dietary components, isoflavones, the gut microbiome, and MetS in hospital employees. Further study is needed to investigate these issues.” Please see the page 11 line 19. |
Reviewer 2 Report
Dear authors,
Your manuscript sending to the IJERPH about the metabolic syndrome in health workers could be of interest to this journal if you can solve the following comments:
- Introduction. Please add more data about the importance of the prevention of Metabolic Syndrome in health workers'. Also, add more data about some risk factors to metabolic syndrome in this population. Explain why is important to know this.
- Materials and methods. Please add the type of study. It is a prospective cohort? It isn't clear. Can you add the bioethical permission to use the data and reproduce it in this manuscript? Please add the data about the selection criteria of this study. For example without a previous diagnosis of metabolic syndrome. Please do a flow chart about the population in the HMIS and this study.
- In this section I have a strong recommendation: You can do a good prospective cohort if you start the follow-up period without any people with metabolic syndrome and follow through the time. After the six years of follow-up, you can do a survival analysis or again repeat this analysis and you can describe the risk factor associated with a higher risk of metabolic syndrome.
- Results. The results in table 2 are obvious, you can describe better other factors associated with metabolic syndrome. Table 3. The year of evaluation will be associated with metabolic syndrome because the chronologic age of the population also increases. So this result could be a confounding factor. Please rewrite the result section.
- Conclusion. You can not say "casual relationship" in an observational study.
Author Response
1
Dear Editor and Reviewers,
Thank you for the opportunity to revise manuscript # ijerph-1324639. We have revised the manuscript based
on the editor and reviewer comments and we wish the quality of the manuscript has improved. Thank you
again for your time and expertise in reviewing this manuscript for publication in Journal of Environmental
Research and Public Health.
Reviewer #2 Comments | Author Response to Comments |
Introduction. Please add more data about the importance of the prevention of Metabolic Syndrome in health workers'. |
Thank you for the comments. We have added the importance of the prevention of MetS in health workers. “According to the trend of the Health Promotion Hospital (HPH), the health of hospital employees is a matter of great concern for hospital management. Hospitals should not only provide health care services to the public and patients but also focus on hospital employees to make the health care. Although hospital employees have better health knowledge, high-pressure work may threaten their health. Moreover, MetS increases individual health care costs by US$3,719, which may have a major impact on the fiscal performance of an institution [1]. Therefore, prevention of MetS is an important issue for hospital employees”. Please see the page 1 line 12. |
Also, add more data about some risk factors to metabolic syndrome in this population. Explain why is important to know this. |
We have added the risk factors to MetS in the hospital employees. " In hospital health care settings, work schedules are commonly characterized by alternating shifts, which make hospital employees particularly vulnerable, as their jobs require them to work night or rotating shifts. The conflicting time signals given by variations in exposure to light at night are proposed to cause the loss of internal synchrony, and the causes of adverse metabolic outcomes have been posited to include a causal link between metabolic regulation and the molecular function of the circadian rhythm [16]. Previous research indicated that individual characteristics, such as shift work [17] is associated with higher rates of MetS." Please see the page 2 line 2. |
Materials and methods. Please add the type of study. It is a prospective cohort? It isn't clear. |
Thank you for the comments. We have checked and clarified that it is a prospective cohort study. Please see the Page 3 line 1. |
Can you add the bioethical permission to use the data and reproduce it in this manuscript? |
We have added the bioethical permission to use the data. ”This study was approved by the Institutional Review Board of the Tri Service General Hospital in Taiwan (approval number: 2-107-05- 030)”. Please see the Page 3 line 2. |
2
Please add the data about the selection criteria of this study. For example without a previous diagnosis of metabolic syndrome. |
Thank you for the comments. We have added the selection criteria of this study. “A total of 1,868 hospital employees completed the health examination in 2012. A total of 1,122 employees who did not complete a total of six years of health check-ups were excluded. A total of 746 hospital employees who completed an annual health check-up during 2012-2017 were included in this study for MetS and non-MetS, and their demographic characteristics, physical characteristics and biochemical profile were compared. The study also measured the association between MetS and demographic information, work type, healthy behaviours, unhealthy behaviours, past history, physical measurements, and biochemical profile by repeated measurement rather than one-shot measurement. Thus, we selected 680 individuals without MetS in 2012 who were enrolled in the analysis for six repeated measurements.” Please see the Page 4 line 12. |
Please do a flow chart about the population in the HMIS and this study. |
We have done a flow chart about the enrollment of the study participants. Please see the figure 1. We revised the manuscript as below. “A total of 1,868 hospital employees completed the health examination in 2012. A total of 1,122 employees who did not complete a total of six years of health check-ups were excluded. A total of 746 hospital employees who completed an annual health check-up during 2012-2017 were included in this study for MetS and non-MetS, and their |
3
demographic characteristics, physical characteristics, and biochemical profile were compared. The study also measured the association between MetS and demographic information, work type, healthy behaviours, unhealthy behaviours, past history, physical measurements, and biochemical profile by repeated measurement rather than one-shot measurement. Thus, we selected 680 individuals without MetS in 2012 who were enrolled in the analysis for six repeated measurements. After six years of follow up, 601 hospital employees were not diagnosed with MetS, and 79 were diagnosed with MetS. The flow chart of the enrolment of study participants is shown in Figure 1”. Please see the Page 4 line 12. |
|
In this section I have a strong recommendation: You can do a good prospective cohort if you start the follow-up period without any people with metabolic syndrome and follow through the time. After the six years of follow-up, you can do a survival analysis or again repeat this analysis and you can describe the risk factor associated with a higher risk of metabolic syndrome. |
Thank you for the recommendation. We have selected 680 hospital employees without metabolic syndrome in 2012 were enrolled in the analysis. After six years of follow-up, 601 hospital employees were not diagnosed with MetS, and 79 were diagnosed with MetS. We re-used the generalized estimating equations (GEEs) analysis to describe the risk factor associated with a higher risk of metabolic syndrome. We ran the data again and showed the results in table 3. |
4
We revised the section as below. “After controlling for sex, shift work, number of chronic diseases, number of family history, smoking and alcohol status, age, BMI, WBC, ALT, and uric acid were significantly associated with a higher risk of MetS. Hospital employees aged between 31 and 40 (OR= 4.596, p=0.009), those aged between 41 and 50 (OR= 7.866, p=0.001), those aged greater than 50 (OR=10.312, p<0.001), those with a BMI 25.0~29.9 kg/m2 (OR=3.934, p<0.001), and those with a BMI ≥ 30 kg/m2 (OR=13.197, p<0.001) had a significantly elevated risk of MetS. Regarding biochemical values, a one-unit increase in WBC count increased the OR for MetS by 19.4% (β=0.177, p=0.001). A one-unit increase in ALT or UA was associated with an increase of 1.3% (β=0.013, p=0.002) and 25% (β=0.223, p=0.005), respectively, in the risk of being diagnosed with MetS.” Please see the Page 7 line 1. |
|
Results. The results in table 2 are obvious, you can describe better other factors associated with metabolic syndrome. |
Thank you for the comments. Other factors associated with metabolic syndrome included the low-density lipoprotein cholesterol (LDL-C) and atherogenic index of plasma (AIP). We have described the factors associated with metabolic syndrome and shown in the table 2. |
5
Because we added the two factors, we revised the sections of methods and results. In the section of material and methods, we introduced the factor of AIP. “AIP was calculated as the logarithmic transformation of the ratio of TG to HDL-C and was calculated according to the following formula: AIP = Log [TGs]/[HDL-C] [24]. AIP values of -0.3 to 0.1 are associated with low risk, 0.1-0.24 with medium risk and above 0.24 with high risk of CVD [25].” Please see the page 4 line 5. In the section of results, we revised as below.” People with MetS had significantly higher weight (t=-10.48, p<0.001), higher WC (t=- 13.13, p<0.001), higher SBP (t=-11.73, p<0.001), higher DBP (t=- 9.24, p<0.001), higher WBC (t=-4.34, p<0.001), higher haemoglobin (t=-3.29, p=0.001), higher FPG (t=-4.48, p<0.001), |
6
higher cholesterol (t=-2.80, p=0.005), higher triglycerides (t=-6.48, p<0.001), higher AST (t=-2.13, p=0.033), higher ALT (t=-3.65, p<0.001), higher total calcium (t=-2.22, p=0.027), higher uric acid (t=-4.87, p<0.001), higher alkaline phosphatase (t=-3.60, p<0.001), lower A/G ratio (t=2.07, p=0.039), lower HDL-C (t=12.75, p<0.001), higher LDL-C (t=-2.69, p=0.008), and higher AIP (t=- 12.96, p<0.001)”. Please see the page 6 line 14. Thanks for reviewer’s recommendation. |
|
Table 3. The year of evaluation will be associated with metabolic syndrome because the chronologic age of the population also increases. So this result could be a confounding factor. Please rewrite the result section. |
Thank you for the recommendation. The year of evaluation could be a confounding factor, so we deleted the variable of “year” and re-used the GEE analysis to describe the risk factors associated with a higher risk of metabolic syndrome. We ran the data again and showed the results in table 3. |
7
We also rewrite the result section. “After controlling for sex, shift work, number of chronic diseases, number of family history, smoking and alcohol status, age, BMI, WBC, ALT, and uric acid were significantly associated with a higher risk of MetS. Hospital employees aged between 31 and 40 (OR= 4.596, p=0.009), those aged between 41 and 50 (OR= 7.866, p=0.001), those aged greater than 50 (OR=10.312, p<0.001), those with a BMI 25.0~29.9 kg/m2 (OR=3.934, p<0.001), and those with a BMI ≥ 30 kg/m2 (OR=13.197, p<0.001) had a significantly elevated risk of MetS. Regarding biochemical values, a one-unit increase in WBC count increased the OR for MetS by 19.4% (β=0.177, p=0.001). A one unit increase in ALT or UA was associated with an increase of 1.3% (β=0.013, p=0.002) and 25% (β=0.223, p=0.005), respectively, in the risk of being diagnosed with MetS.” Please see the Page 7 line 1. |
|
Conclusion. You can not say "casual relationship" in an observational study. |
Thank you for the comments. We have deleted the “causal relationship”. We addressed as below. “Our study improves the understanding of MetS among hospital employees with data from a long-term follow-up hospital health examination. The results of the longitudinal study design showed that increasing age, higher BMI, and higher WBC, ALT, and UA levels were factors that had a significantly elevated risk of MetS over the six years.” Please see the Page 12 line 13. |
Round 2
Reviewer 1 Report
The authors have satisfactorily responded to all my questions and made the necessary changes to the manuscript.
Reviewer 2 Report
Dear authors,
Thanks for your work to improve the manuscript. But please add the "Figure 1" there is not in the manuscript.
Please review the languages with a native speaker.